# Selection of Operation Conditions for a Batch Brown Seaweed Biosorption System for Removal of Copper from Aqueous Solutions



Henrik K. Hansen [1],*, Claudia Gutiérrez [1], Natalia Valencia [1], Claudia Gotschlich [1], Andrea Lazo [1], Pamela Lazo [2] and Rodrigo Ortiz-Soto [3]

1 Departamento de Ingeniería Química y Ambiental, Universidad Técnica Federico Santa María, Avenida España 1680, Valparaíso 2390123, Chile; claudia.gutierrez@usm.cl (C.G.); natalia.valencia@alumnos.usm.cl (N.V.); claudia.gotschlich@gmail.com (C.G.); andrea.lazo@usm.cl (A.L.)
2 Instituto de Química y Bioquímica, Facultad de Ciencias, Universidad de Valparaíso, Valparaíso 2362735, Chile; pamela.lazo@uv.cl
3 Escuela de Ingeniería Química, Pontificia Universidad Católica de Valparaíso, Valparaíso 2340025, Chile; rodrigo.ortiz@pucv.cl
* Correspondence: henrik.hansen@usm.cl

**Abstract:** Heavy metal exposure from wastewater is an important environmental issue worldwide. In the search for more efficient treatment technologies, biosorption has been presented as an alternative for contaminant removal from wastewaters. The aim of this work is to determine the operation parameters of copper adsorption followed by biosorbent regeneration. The algae *Durvillaea antarctica* and *Lessonia trabeculata* were used as biosorbents in batch experiments. These biosorbents were exposed to different conditions, such as pH, copper concentration, exposure time, mass-to-volume ratios and regeneration reagents. Batch sorption tests revealed an adequate pH of 4.5–5.0. The selected biosorbent was *D. antarctica* due to a considerably higher copper retention capacity. As a regenerating reagent, sulfuric acid was more efficient. For diluted copper solutions (10 to 100 mg L$^{-1}$), a biosorbent particle size of between 1.70 and 3.36 mm showed better retention capacity than larger particles and a biosorbent mass-to-volume ratio of 10 g L$^{-1}$ was desirable for these metal concentrations.

**Keywords:** copper retention; biomass particle size; sorption isotherms; sorption kinetics





## 1. Introduction

Water resources, which are vital for life, have been reduced partly due to contamination mainly caused by industrial processes, agriculture and urbanization. In Chile, mining is the most relevant productive industrial activity, so focus has to be turned onto the waste generated by this kind of activity [1–4]. Mining, besides being the largest production activity of the country, is also the principal cause of heavy metal contamination.

Within great mineral reservoirs in Chile, the production of copper, iron, molybdenum, lead, zinc, gold and silver is considered, with copper and molybdenum (a byproduct of copper production) being the most interesting ones. To obtain these metals, large water consumption is required for use in different operations, such as extraction, grinding, concentration and refinement. Once used, the wastewater contains a high amount of heavy metals.

The toxicity of heavy metals and their effect on the environment has created a need to reduce their concentration in industrial effluents below the levels required by environmental legislation. This has initiated a search for alternative methods for the elimination of these elements from aqueous solutions.

Normally, heavy metals are removed by physicochemical treatment methods such as chemical precipitation, reverse osmosis, adsorption in activated carbon, electrodialysis, and ion exchange, but it has to be noted that these processes are expensive and could be inefficient. Because of this, biosorption is being studied as a heavy metal removal

technique, which is a promising technology for wastewater treatment with low concentrations (1–100 mg $L^{-1}$). Biosorbent material can be regenerated several times. It is found in abundance in the environment and is an inexpensive resource that is easy to access; often biomass waste is used [5–8].

A biosorption process refers to the ability of materials of biological origin to retain heavy metals from diluted aqueous solutions in their structures [9]. It is considered a clean environmental remediation process for metal recovery and decontamination of wastewater by heavy metals and metalloids, such as copper, lead, cadmium, nickel and arsenic [10]. Biomass traditionally used in biosorption processes belongs to three groups: bacteria, fungi or algae. In relation to its source, it can be obtained directly from nature or as a waste from productive processes [11] (Franco et al., 2021).

Seaweeds (or marine algae) have previosly been reported to be efficient in inorganic contaminant removal from wastewater [12,13]. In brown seaweeds, fibers are mainly cellulose and insoluble alginates [14]. These alginates are Ca, Mg, or Na salts of alginic acid (1,4-linked polymer of β-D-mannuronic acid and α-L-guluronic acid). Alginates are known for their high divalent metal cation uptake capacities [15] and are therefore a suitable adsorbent for copper removal. Two brown seaweed species that are abundant along the entire coastline of Chile contain significant amounts of alginates. In *D. antarctica*, typically between 10–20% and sometimes even around 50% of the total dry weight are alginates [16,17], whereas *L. trabeculata* was reported to contain similar amounts of alginates [18]. Therefore, both seaweed species would be possible metal cation accumulaters when treating aquéuos solutions with adsorption processes using these biosorbents. Furthermore, these seaweeds are actually so abundant in Chile that they appear as solid waste when cleaning the beaches and coast—meaning that they provide very low-cost sorbent material.

In Chile, both mineral processing wastewater and acidic mine drainage contain copper in concentrations that would be favorable for biosorption [19]. Until now, no copper uptake data have been published with regard to *L. trabeculata*, and only Cid et al. (2015) [20] have investigated this the behavior of *D. antarctica* under some specific conditions with respect to pH, biosorbent mass-to-solution volume, copper concentration and particle size, so it would be interesting to compare these biosorbents with previously reported copper retentions for other sorbents, in particular seaweeds, that would be more difficult to have access to in Chile. Table 1 shows a summary of research studies on the copper retention of a variety of brown seaweeds. Furthermore, to optimize the applicablity of the sorbents in a real treatment process, it would be necessary to evaluate the possiblity to regenerate the biosorbents, so that (1) biosorbent disposal would be minimized and (2) copper could be recovered.

**Table 1.** Comparison of maximum copper biosorption capacities ($q_{max}$) of different brown seaweeds.

| Brown Seaweed | Cu conc. (mg $L^{-1}$) | pH | Particle Size (mm) | Biomass/ Volumen (g/L) | Time to Equilibrium (hours) | Temperature (°C) | $q_{max}$ (mg $g^{-1}$) | Reference |
|---|---|---|---|---|---|---|---|---|
| *Lessonia nigrescens* | 7.5–300 | 5 | 0.5–1 | 1 | 2 | 20 | 60.4 | [21] |
| *Cystoseira* sp. | 10–30 | 6 | <0.5 | 0.1 | 2 | 28 | 180.4 | [22] |
| *Lessonia nigrescens* blades | 200–1000 | 3.2 | 5–20 | 1 / 4 | 168 | 25 | 56.2 / 47.3 | [7] |
| *Lessonia nigrescens* stipes | 200–1000 | 3.2 | 10–15 | 1 / 4 | 168 | 25 | 78.8 / 218.7 | [7] |
| *Sargassum tenerrimum* | 10–50 | 5 | 0.2–0.5 | 10 | 24 | 28 | 39.8 | [23] |
| *Iyengaria stellata* | 10–50 | 5 | 0.2–0.5 | 10 | 24 | 28 | 46.3 | [23] |
| *Lobophora variegata* | 10–50 | 5 | 0.2–0.5 | 10 | 24 | 28 | 38.0 | [23] |
| *Cystoseira indica* | 10–50 | 5 | 0.2–0.5 | 10 | 24 | 28 | 30.9 | [23] |
| *Sargassum cinereum* | 10–50 | 5 | 0.2–0.5 | 10 | 24 | 28 | 34.0 | [23] |
| *Durvillaea antarctica* | 7.5–300 | 5 | 0.5–1 | 1 | 2 | 20 | 91.5 | [20] |

**Table 1.** *Cont.*

| Brown Seaweed | Cu conc. (mg L$^{-1}$) | pH | Particle Size (mm) | Biomass/ Volumen (g/L) | Time to Equilibrium (hours) | Temperature (°C) | q$_{max}$ (mg g$^{-1}$) | Reference |
|---|---|---|---|---|---|---|---|---|
| *Sargassum filipendula* | 19–265 | 4.5 | 0.855 | - | - | 20 | 84.1 | [24] |
| *Sargassum sinicola* | 2–256 | - | 0.2–0.5 | 10 | 24 | - | 116.6 | [25] |
| *Fucus serratus* | 0.6–25 | 5.5 | 0.355–0.5 | 0.09 | 8 | 20 | 101.8 | [26] |
| *Fucus vesiculosus* | 10–150 | 5 | <0.5 | 0.5 | 2 | 23 | 105.5 | [27] |
| *Sargassum* sp. | 20–500 | 5.5 | 0.5 | 1 | 3 | 22 | 72.5 | [28] |
| *Sargassum* sp. | - | 6 | <0.325 | 2 | 4 | 22 | 84.0–86.9 | [29] |
| *Fucus spiralis* | 10–150 | 4 | <0.5 | 0.5 | 2 | - | 70.9 | [30] |
| *Ascophyllum nodosum* | 10–150 | 4 | <0.5 | 0.5 | 2 | - | 58.8 | [30] |
| *Sargassum* sp. | - | 5 | 0.5–0.8 | 1 | 6 | 22 | 62.9 | [31] |
| *Padina* sp. | - | 5 | 0.5–0.8 | 1 | 6 | 22 | 72.4 | [31] |
| *Sargassum vulgare* | 10–250 | 4.5 | 1–4 | 2 | 6 | 22 | 59.1 | [32] |
| *Sargassum fluitans* | 10–150 | 4.5 | 1–4 | 2 | 6 | 22 | 50.8 | [32] |
| *Sargassum filipendula* | 10–250 | 4.5 | 1–4 | 2 | 6 | 22 | 56.6 | [32] |

In this part of the research work, a batch copper removal process is developed, using the algae *D. antarctica* and *L. trabeculata* as biosorbents. These biosorbents are available along the coast of Chile. The specific objectives of the study are (i) to choose an adequate operating pH, (ii) to choose the appropriate biosorbent for a continuous process, (iii) to choose particle size, (iv) to choose an adequate mass of biosorbent/volume of solution ratio, (v) to study adsorption kinetics, (vi) to study adsorption isotherm and (vii) to choose the regeneration reagent that recovers the most copper from the biosorbent.

## 2. Materials and Methods

Based on biosorption experiments, it is possible to quantitatively assess the retention capacity of a biosorbent by using a solution with a specific contaminant. For the evaluation of the retention capacity, a simple metal mass balance is used, which follows the logical assumption that the metal ion loss in the solution is the metal retained by the biosorbent, as shown in Equation (1).

$$q = \frac{V \cdot (C_i - C_{eq})}{M} \tag{1}$$

where $C_i$ (mg L$^{-1}$) is the initial concentration of the element in the solution, V (L) is the initial solution volume, $C_{eq}$ (mg L$^{-1}$) is the equilibrium concentration of the element in solution, M (g) is the biosorbent mass, and q (mg g$^{-1}$) is the retention capacity of the element by the biosorbent.

The biosorption phenomena are time dependent, thus it is necessary to obtain the adsorption rate for the design and evaluation of a potential biosorbent. Furthermore, the fitting of both biosorption kinetic and equilibrium data with conventional mathematical models would enlighten the efficiency of the metal uptake. Table 2 summarizes the models used in this work.

### 2.1. Reagents

The copper solutions were prepared by dissolving CuSO$_4$·5H$_2$O 99.5% (analytical grade) in distilled water. pH was adjusted by the addition of hydrochloric acid 37% GR for analysis (Merck, Rahway, NJ, USA) or by the addition of sodium hydroxide NaOH (5 M) prepared by dissolving 98% extra pure sodium hydroxide pellets (Loba Chemie, Mumbai, India).

**Table 2.** Mathematical biosorption models for kinetic and isotherm data.

| Model Type | Equation | Parameter Description |
|---|---|---|
| Kinetic model | | |
| Pseudo first order Lagergren | $q_t = q_{eq}\left(1 - e^{-k_{ad}t}\right)$ | $q_t$ (mg g$^{-1}$) is the adsorbate retention in time t, $q_{eq}$ (mg g$^{-1}$) is the adsorbate retention in equilibrium, $k_{ad}$ (min$^{-1}$) is the adsorption first order constant and t (min) is the time. |
| Pseudo second order Ho & McKay | $q_t = \dfrac{t}{\dfrac{1}{k \cdot q_{eq}^2} + \dfrac{t}{q_{eq}}}$ | k (g mg$^{-1}$ min$^{-1}$) is the second order adsorption constant. |
| Isotherm model | | |
| Freundlich | $q_{eq} = k \cdot C_{eq}^{1/n}$ | k is the Freundlich capacity parameter and 1/n is the Freundlich intensity parameter. |
| Langmuir | $q_{eq} = \dfrac{q_m \cdot b \cdot C_{eq}}{1 + b \cdot C_{eq}}$ | $q_m$ is the maximum concentration of the metal on the biomass (mg metal g$^{-1}$ dry biosorbent), b is a coefficient related to the affinity between the biosorbent and the metal, high values of b indicate a high affinity for the biosorbent and show a steep initial slope in the isotherm plot (L mg$^{-1}$). |
| Sips | $q_{eq} = \dfrac{q_m \left(K_S \cdot C_{eq}\right)^{1/n_s}}{1 + \left(K_S \cdot C_{eq}\right)^{1/n_s}}$ | $K_S$ (L mg$^{-1}$) is the equilibrium constant and $n_s$ (-) is the model exponent. |
| Brunauer, Emmett and Teller | $q_e = \dfrac{q_m k_1 C_e \left[1 - (n+1)(k_2 C_e)^n + n(k_2 C_e)^{n+1}\right]}{(1 - k_2 C_e)\left[1 + \left(\dfrac{k_1}{k_2} - 1\right)k_2 C_e - \left(\dfrac{k_1}{k_2}\right)(k_2 C_e)^{n+1}\right]}$ | $q_m$ is the maximum adsorbate retention in the monolayer (mg g$^{-1}$), $k_1$ is the equilibrium constant of adsorption in the first layer (L mg$^{-1}$), $k_2$ is the equilibrium constant of adsorption in upper layers (L mg$^{-1}$) and n is the number of adsorption layers estimated. |

### 2.2. Analytical

Each liquid sample was filtered through a N° 131 grade filter paper (Advantec, Dublin, CA, USA) by a vacuum pump (Welch, Ilmenau, Germany—model 2522). The copper concentration in the filtrate was determined by atomic absorption spectrophotometry in flame (Varian, Palo Alto, CA, USA—model SpectrAA 55) according to Chilean standard NCh 2313/10 Of. 96. pH was measured using an Orion (Thermo Scientific, Beverly, MA, USA) PerpHect logR model 370 pH meter with a combined pH electrode.

### 2.3. Preparation of Adsorbent

*D. antarctica* and *L. trabeculata* samples were collected in the bay of Valparaíso, Chile. After sampling, the algae were washed in tap water and then in distilled water to remove any salt present. The algae were dried at 50 °C until they obtained a constant weight. The dry biosorbents were first cut with a knife into regular-shaped pieces and then a jaw crusher was used to obtain the smaller-sized particles. A RO-TAP Sieve Shaker with test sieves from W.S. Tyler, model RX-29-10, was used to obtain different size fractions. The particle size ranges separated by sieving and chosen for the experiments were: 0.43–1.70 mm, 1.70–3.36 mm, 3.36–4.00 mm and 4.00–5.66 mm.

### 2.4. Experimental Plan

The conditions of every experimental run are summarized in Table 3. The analyzed parameters were: (a) operation pH, (b) algae used as biosorbent, (c) regenerating reagent, (d) biosorbent particle size, (e) biosorbent mass-to-solution volume ratio (M/V), (f) time sorption and (g) adsorption isotherm.

**Table 3.** Summary of experimental details.

| Experimental Run | Cu conc. mg L$^{-1}$ | M/V Ratio g L$^{-1}$ | Particle Size mm | Time min | Biosorbent | pH |
|---|---|---|---|---|---|---|
| pH determination | 2 | 10 | 4.00–5.66 | 30 60 120 | *D. Antarctica* *L. trabeculata* | 3.0–3.5 4.5–5.0 |
| Biosorbent determination | 30 100 | 10 | 4.00–5.66 | 30 60 120 300 1440 | *D. Antarctica* *L. trabeculata* | 4.5–5.0 |
| Regenerating reagent determination | 100 | 10 | 4.00–5.66 | 10 20 30 120 | *D. antarctica* | 4.5–5.0 |
| Biosorbent particle size | 10 100 | 20 | 4.00–5.66 3.36–4.00 1.70–3.36 0.43–1.70 | 360 | *D. antarctica* | 4.5–5.0 |
| M/V Ratio | 10 100 | 10 20 40 | 1.70–3.36 | 1440 | *D. antarctica* | 4.5–5.0 |
| Biosorption kinetics | 10 100 | 10 | 1.70–3.36 | 5 10 20 30 60 120 360 720 | *D. antarctica* | 4.5–5.0 |
| Adsorption isotherm | 10 25 50 75 100 | 10 | 1.70–3.36 | 360 | *D. antarctica* | 4.5–5.0 |

The pH in the solution was kept constant by adding drops of either 0.5 M HCl or 0.5 M NaOH solutions, assuring that the total liquid volume was not affected severely. The experiments were carried out in duplicate, without stirring and at an ambient temperature (20–25 °C). Relative standard deviations were lower than 5% in every experiment. The standard deviations and error margins are given in the tables and figures representing the experimental results.

## 3. Results

### 3.1. Determination of Sorption pH

One of the most relevant factors in metal ion retention with seaweeds is pH [33–35]; therefore, experiments were carried out at two pH intervals, 3.0–3.5 and 4.5–5.0, in the solutions. Lower pH values were not analyzed because it is generally known that the sorption is worse [31,36,37], and higher pH values were not chosen to avoid the effect of chemical precipitation.

For these experiments, 2 mg L$^{-1}$ copper solutions were used, from which 500 mL was poured into every beaker and 5 g of dried *L. trabeculata* or *D. antarctica* was added in each case. After the treatment time was reached, the solution was filtered and the copper content in the liquid was measured. Experimental results for the copper retention capacity of *L. trabeculata* and *D. antarctica* at different times for pH intervals of 3.0–3.5 and 4.5–5.0, respectively, are displayed in Figure 1.

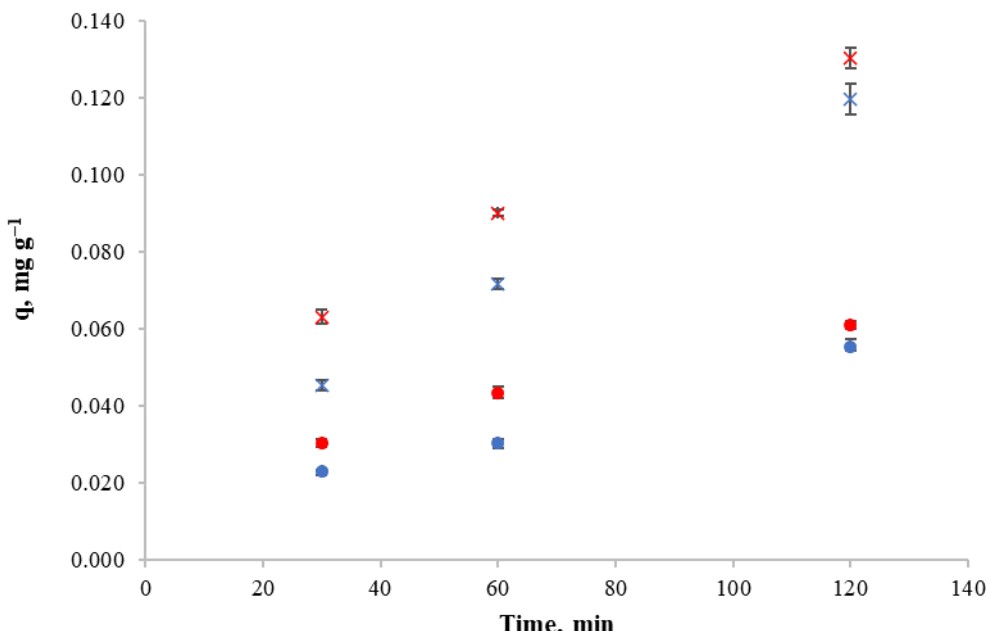

**Figure 1.** Copper retention capacity in time of different algae and pH. (•): *Lessonia trabeculata* at pH of 3.5–4.0. (•): *Lessonia trabeculata* at pH of 4.5–5.0. (×): *Durvillaea antarctica* at pH of 3.5–4.0. (×): *Durvillaea antarctica* at pH of 4.5–5.0.

In Figure 1, the increase in retention capacity of *L. trabeculata* and *D. antarctica* by time can be observed for both pH intervals. Furthermore, at the highest pH, a higher retention capacity is obtained at every time. Thus, the operation pH for the following experiments is 4.5–5.0.

*3.2. Biosorbent Determination*

In order to maximize the biosorption process, the highest copper retention has to be achieved in the minimum contact time, when focusing on continuous systems. For this reason, experiments at different times were carried out with *L. trabeculata* and *D. antarctica* as biosorbents, with the objective of choosing the most effective biosorbent.

For these experiments, 100 mg L$^{-1}$ and 30 mg L$^{-1}$ copper solutions were prepared, from which 500 mL was poured into every beaker and 5 g of dried *L. trabeculata* or *D. antárctica*, as it corresponds, was added in each case. After the treatment time was reached, the solution was filtered and the copper content in the liquid phase was measured. Experimental results for the copper retention capacity of both biosorbents at different times are shown in Figure 2.

In Figure 2, it can be noticed that from the beginning, *D. antarctica* has a considerably higher copper retention than *L. trabeculata* for both initial metal concentrations. This can be assumed because at 100 mg L$^{-1}$ of copper initial concentration, the metal mass retention capacity of *D. antarctica* is, on average, 82% higher than that of *L. trabeculata*. For the initial copper concentration of 30 mg L$^{-1}$, the metal mass retention capacity of *D. antarctica* is, on average, 46% higher than *L. trabeculata*.

This can be explained by the alginate content of the algae, which in *D. antarctica* could be as high as in the range of between 30 and 55% d.wt., and in *L. trabeculata*, it is in the range of 15–21% d.wt. [17,38,39]. Alginates are responsible for the strong affinity that the algae show for heavy metals such as copper [40,41].

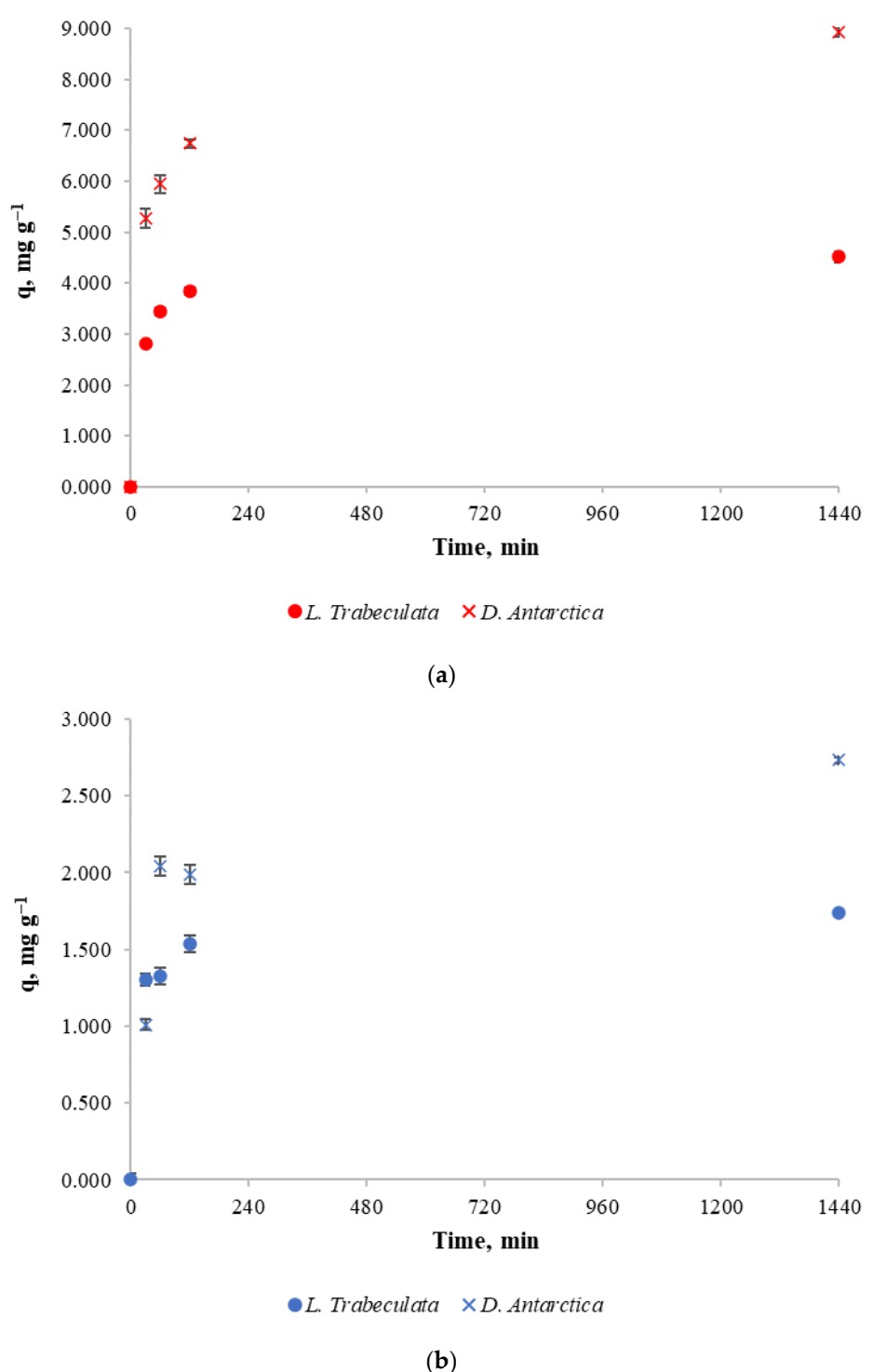

**Figure 2.** Copper retention capacity of different algae and copper initial concentrations. (**a**): (●): *Lessonia trabeculata* and copper initial concentration of 100 mg L$^{-1}$. (×): *Durvillaea antarctica* and copper initial concentration of 100 mg L$^{-1}$. (**b**): (×): *Durvillaea antarctica* and copper initial concentration of 30 mg L$^{-1}$. (●): *Lessonia trabeculata* and copper initial concentration of 30 mg L$^{-1}$.

### 3.3. Biosorbent Particle Size Determination

Experiments for determining the effect of biosorbent particle size in copper removal from the solution were carried out with the aim of obtaining the most copper removed, using *D. Antarctica* as the biosorbent with copper solutions of 10 and 100 mg L$^{-1}$, respec-

tively, and a contact time of 360 min. Experimental results for copper retention capacity of every particle size for different initial copper concentrations are shown in Table 4.

**Table 4.** Biosorbent particle size determination experimental runs.

| Experimental Run | Particle Size Range mm | Initial Cu Concentration mg $L^{-1}$ | |
|---|---|---|---|
| | | **10** | **100** |
| | | Retention Capacity mg $g^{-1}$ | |
| $T_4$ | 4.00–5.66 | $0.371 \pm 0.014$ | $2.372 \pm 0.031$ |
| $T_3$ | 3.36–4.00 | $0.325 \pm 0.013$ | $2.531 \pm 0.044$ |
| $T_2$ | 1.70–3.36 | $0.358 \pm 0.013$ | $2.681 \pm 0.051$ |
| $T_1$ | 0.43–1.70 | $0.312 \pm 0.015$ | $2.657 \pm 0.055$ |

It can be noticed that for the initial copper concentration of 10 mg $L^{-1}$, the particle size corresponding to $T_4$ and $T_2$ experimental runs presented the best retention capacity. For an initial copper concentration of 100 mg $L^{-1}$, the best results were in the $T_1$ and $T_2$ experimental runs. Thus, a biosorbent particle size of 1.70 to 3.36 mm was selected for further experiments.

*3.4. Mass/Volume Ratio Determination*

In order to determine the biosorbent mass-to-solution volume ratio, 10 and 100 mg $L^{-1}$ of initial copper concentrations and a contact time of 24 h were used so that the influence of biosorbent mass/volume ratio in retention capacity and metal removal could be observed by using 10, 20 and 40 g of biosorbent per liter of solution.

Experimental results for copper retention capacity and copper removal against different mass/volume ratios for both initial copper concentrations are shown in Figure 3. The decrease in copper retention capacity of the biosorbent as the mass/volume ratio increases can be observed for both initial concentrations. This indicates that for those concentrations, it is not beneficial to increase the biosorbent concentration. Thus, the mass/volume ratio of 10 g $L^{-1}$ achieves high retention capacity.

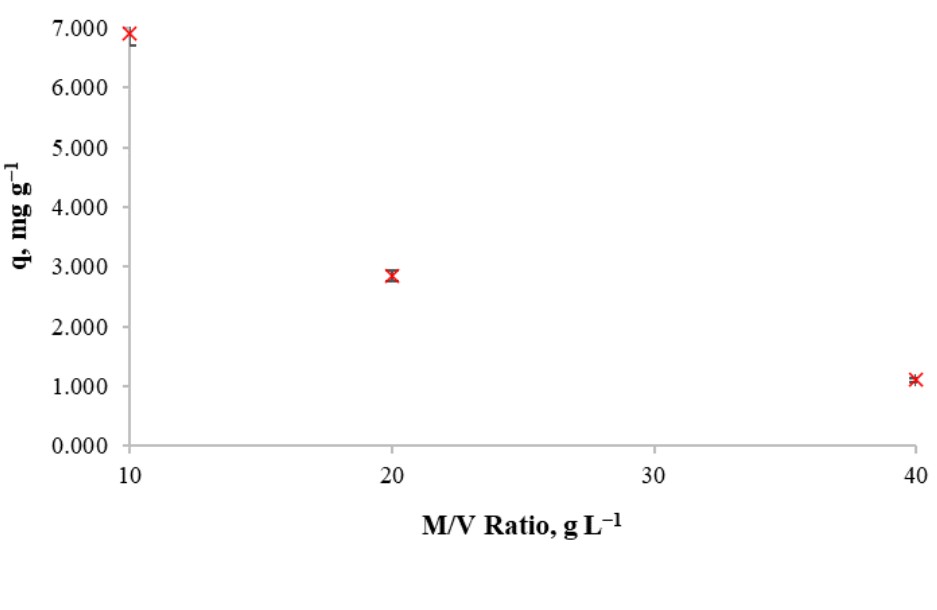

**(a)**

**Figure 3.** *Cont.*

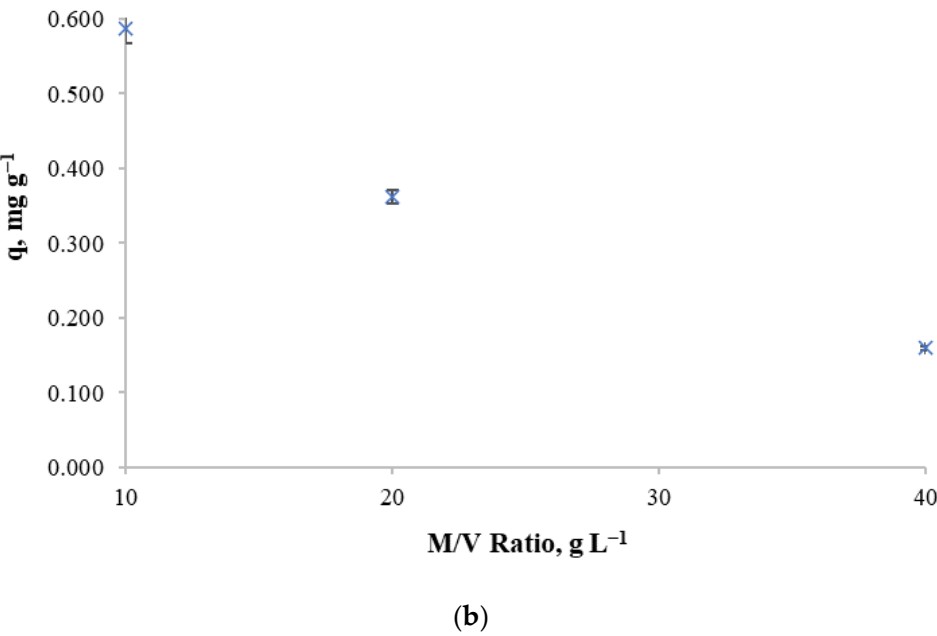

(**b**)

**Figure 3.** Copper retention capacity at different mass/volume ratio for (**a**): an initial copper concentration of 100 mg L$^{-1}$, (**b**) an initial copper concentration of 10 mg L$^{-1}$.

### 3.5. Biosorption Kinetics

In order to determine biosorption kinetics, a 500 mL copper solution with an initial concentration of 10 and 100 mg L$^{-1}$ and a pH of 4.5–5.0 was used with 5 g of biosorbent mass. Experiments were based on a change in contact time, which was between 5 and 720 min. After that time, the biosorbent was withdrawn from the solution and the copper concentration in the solution was analyzed.

Experimental results for copper retention capacity against contact time and best fits for the Lagergren and Ho & Mckay models are shown in Figure 4. According to the figure, it can be noticed that the copper retention by the biosorbent increases considerably during the first minutes of contact until equilibrium is achieved at 360 min for both cases. The retention capacity does not increase significantly after that time due to process stabilization.

Experimental data was fitted to both the Lagergren and Ho & Mckay models and the obtained parameters for each model are presented in Table 5. It can be concluded from the determination coefficient $R^2$ that the Ho & Mckay model fits the experimental data better than the Lagergren model when a mass/volume ratio of 10 g L$^{-1}$ and initial copper concentration of 100 mg L$^{-1}$ is used.

**Table 5.** Lagergren and Ho & Mckay model parameter values.

| Model | Initial Cu Concentration mg L$^{-1}$ | | Model | Initial Cu Concentration mg L$^{-1}$ | |
|---|---|---|---|---|---|
| | 10 | 100 | | 10 | 100 |
| Ho & Mckay | | | Lagergren | | |
| q$_{eq}$ mg g$^{-1}$ | 0.585 ± 0.009 | 6.513 ± 0.077 | q$_{eq}$ mg g$^{-1}$ | 0.589 ± 0.005 | 6.202 ± 0.041 |
| k g mg$^{-1}$ min$^{-1}$ | 0.076 ± 0.002 | 0.035 ± 0.001 | k$_{ad}$ min$^{-1}$ | 0.024 ± 0.001 | 0.145 ± 0.002 |
| $R^2$ | 90.6% | 95.8% | $R^2$ | 81.6% | 89.9% |

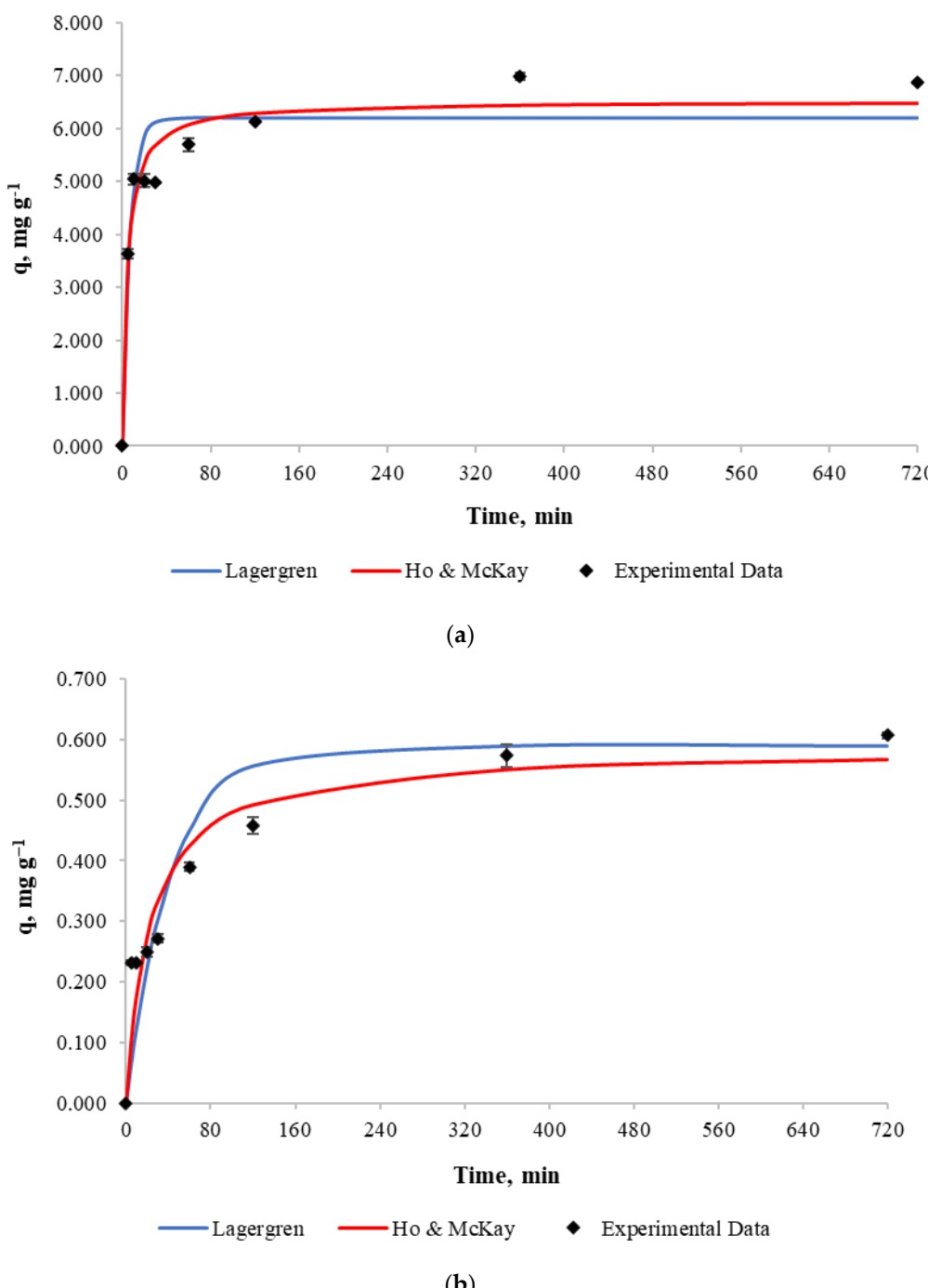

**Figure 4.** Biosorption kinetics (**a**): Initial copper concentration of 100 mg L$^{-1}$. (**b**): Initial copper concentration of 10 mg L$^{-1}$.

### 3.6. Adsorption Isotherm Determination

To determine the adsorption isotherms, experiments with 10, 25, 50, 75 and 100 mg L$^{-1}$ copper concentration, a mass/volume ratio of 10 g L$^{-1}$ and the same other conditions of the other experimental runs were carried out. Each mathematical model shown in Table 2 was used for adsorption isotherm determination that relates to the amount of copper adsorbed by the algae (retention capacity) and the equilibrium concentration in the solution.

To determine the parameters of the Freundlich, Sips and BET models, the Microsoft Excel SOLVER tool was used for data optimization, which uses the minimum squares error method. On the other hand, the Langmuir model parameters determination was made by using the linearization of the model, but in this case the parameter values were negative,

so it can be concluded that the Langmuir model does not represent the process for the equilibrium concentrations used.

Experimental results for copper equilibrium retention capacity against copper equilibrium and the fitted Freundlich, Sips, BET and Langmuir adsorption isotherms are shown in Figure 5. Parameters and representative statistical values of the four models are summarized in Table 6.

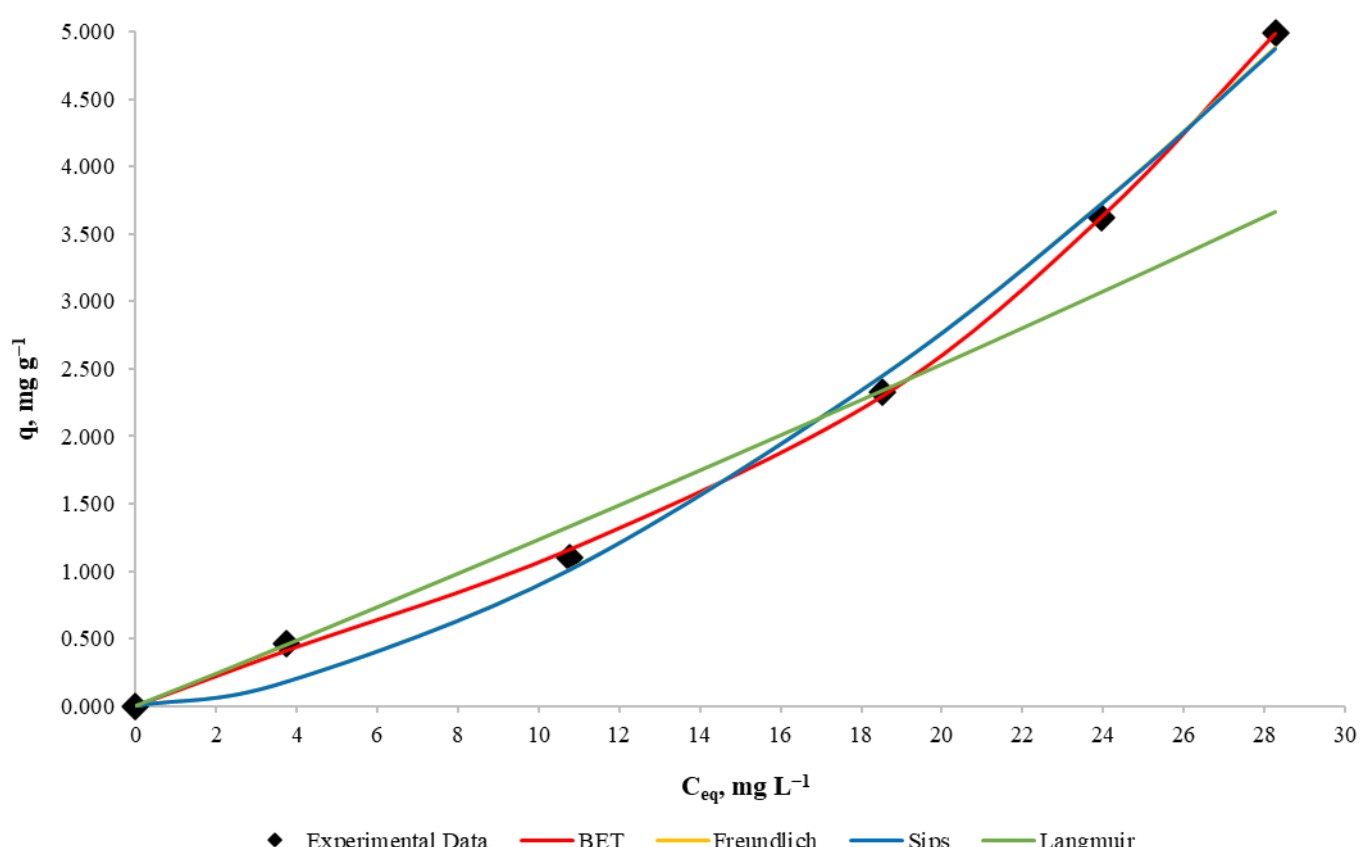

**Figure 5.** Adsorption isotherms for Cu biosorption with *D. Antarctica*. Cu equilibrium biomass uptake as a function of solution concentration at equilibrium.

**Table 6.** Isotherm model parameter values.

| Model | Parameters (Units in Table 2) | | Residuals Sum of Squares | Determination Coefficient $R^2$ |
|---|---|---|---|---|
| Freundlich | k | 0.021 | $1.247 \times 10^{-1}$ | 99.34% |
| | n | 0.613 | | |
| Langmuir | $q_m$ | −47.29 | 2.115 | 88.75% |
| | b | −0.00254 | | |
| Sips | $q_m$ | 623.4 | $1.247 \times 10^{-1}$ | 99.33% |
| | $K_S$ | $3.303 \times 10^{-5}$ | | |
| | $n_S$ | 1.638 | | |
| BET | $q_m$ | 3.955 | $7.010 \times 10^{-3}$ | 99.96% |
| | $k_1$ | $2.969 \times 10^{-2}$ | | |
| | $k_2$ | $2.821 \times 10^{-2}$ | | |
| | n | 11.08 | | |

From visual analysis of Figure 5, it can be noticed that the Langmuir model fits the data poorly, even when using its obtained parameters. On the other hand, the Freundlich model fits the data as well as the Sips model, and because their differences are not possible to detect, it is deduced that the Sips model is an overparameterization in this case. It can be noticed that the BET model has the best resemblance to the data, even better than the Freundlich model, which supports the idea that for the experimental conditions, the biosorbent is not saturated, so it can be used for more concentrated solutions under these conditions. In terms of model selection, the BET model presents a better determination coefficient than the Freundlich model (99.96% for BET and 99.34% for Freundlich), which could be assumed as negligible, but after performing a Fisher statistical test for model comparison [42], the *p*-value was below 3%. Therefore, there is statistical evidence that supports the BET model as the best fit for the analyzed experiment, so these algae can form multiple adsorption layers and have a monolayer adsorption capacity of 3.955 mg g$^{-1}$.

The maximum retention capacity should only be taken as an indicator of whether or not the biosorbent would be useful because the maximum retention capacity is never reached in an actual sorption wastewater treatment plant. Therefore, the retention capacities should be supplemented by the kinetic phenomena of the biosorption in order to estimate a decent residence time in a treatment process [10].

### 3.7. Regenerating Reagent Selection

This series of experiments was carried out for the selection of the regenerating reagent that removes the highest quantity of copper from the biosorbent. Among the regenerating reagent requirements, there should be (i) a high copper affinity, (ii) maintenance of biosorbent properties after contact, (iii) easy access, and (iv) low cost. Thus, the analyzed regenerating reagents were sulfuric acid and hydrochloric acid.

The results of this experiment aim to enhance the diffusion of copper from biosorbent to regenerating reagent because of its affinity to copper; therefore, the biosorbent holds a low copper concentration and is able to be reused for copper biosorption.

For assessing copper affinity with regenerating reagent, a copper sulfate solution was prepared, with 100 mg L$^{-1}$ of copper as in the previous experiments, from which 500 mL was poured into every beaker and was in contact with 5 g of dried *D. antarctica* for 24 h. Then the biosorbent was rinsed twice with distilled water and 0.1 mol L$^{-1}$ (pH 1) sulfuric acid or 0.1 mol L$^{-1}$ (pH 1) hydrochloric was added for a determined time. After that time, the biosorbent was withdrawn from the solution and the amount of desorbed copper was determined by the difference of adsorbed copper mass from the first solution and desorbed mass from the second solution. Experimental results for copper desorption when applying H$_2$SO$_4$ and HCl with different contact times are shown in Figure 6.

It can be observed that for both regenerating reagents, copper is re-adsorbed in the biosorbent as contact time increases. In case of hydrochloric acid, desorption decreases drastically at 20 min, because copper is re-adsorbed rapidly, but in the case of sulfuric acid, the copper re-adsorption is lower and slower than hydrochloric acid desorption.

Because sulfuric acid (a) shows a more stable performance with copper as sorbate and *D. antarctica* as biosorbent than hydrochloric acid, (b) is widely available and (c) has a higher purity than hydrochloric acid, it is recommended as a regenerating reagent.

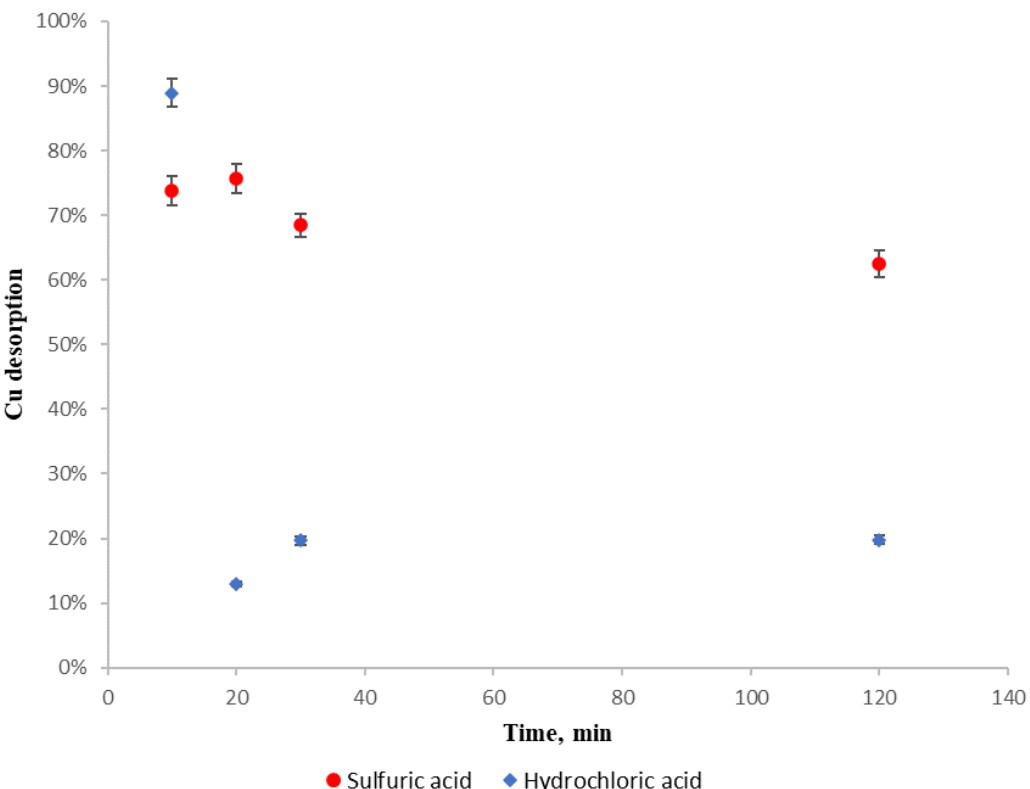

**Figure 6.** Regenerating reagents desorption capacity. (●): $H_2SO_4$. (♦): HCl.

## 4. Conclusions

From the results, it can be concluded that the brown seaweeds *Lessonia trabeculata* and *Durvillaea antarctica* have an important copper retention capacity in acidic solutions, where the capacity of *D. antarctica* is twice as high as that of *L. trabeculata*. Thus, *D. antarctica* is more recommended for continuous systems. Heavy metal removal efficiency varies with pH. From the analysis of two intervals, a pH between 4.5 and 5.0 gives better results in copper removal with the studied biosorbents.

For process kinetic parameters determination applying the Lagergren and Ho & McKay models, it is concluded that the Ho & McKay model fits the experimental data better. Concerning adsorption isotherms, the BET model shows the best fit, indicating that the biosorbent is not saturated. The chosen regenerating reagent is sulfuric acid as it presents higher copper removal values and shows no sign of metal re-adsorption before 30 min, whereas hydrochloric acid shows copper re-adsorption after 10 min.

**Author Contributions:** Conceptualization, H.K.H., C.G. (Claudia Gutiérrez), N.V. and C.G. (Claudia Gotschlich); methodology, N.V. and C.G. (Claudia Gotschlich); validation, H.K.H., A.L. and P.L.; formal analysis, H.K.H., C.G. (Claudia Gutiérrez) and R.O.-S.; investigation, N.V. and C.G. (Claudia Gotschlich); writing—original draft preparation, H.K.H., C.G. (Claudia Gutiérrez) and R.O.-S.; writing—review and editing, H.K.H., C.G. (Claudia Gutiérrez) and R.O.-S.; visualization, H.K.H., C.G. (Claudia Gutiérrez), A.L., P.L. and R.O.-S.; supervision, H.K.H. All authors have read and agreed to the published version of the manuscript.

**Funding:** This research was funded by the Chilean National Research Foundation (ANID) Fondecyt de Iniciación en Investigación, grant N° 11200189 and Fondecyt Regular, grant N° 1220712. The APC was funded by Metals.

**Institutional Review Board Statement:** Not applicable.

**Informed Consent Statement:** Not applicable.

**Data Availability Statement:** The data presented in this study are available on request from the corresponding author.

**Conflicts of Interest:** The authors declare no conflict of interest.

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
