# Peer review of "Selection of Operation Conditions for a Batch Brown Seaweed Biosorption System for Removal of Copper from Aqueous Solutions"

_metals, doi:10.3390/met13061008_

Round 1

Reviewer 1 Report

In this paper, authors used the algae Durvillaea antarctica and Lessonia trabeculata as copper biosorbents in batch experiments. The aim was to determine the operation parameters (pH, copper concentration, exposure time, mass-to-volume ratios and regeneration reagents) of adsorption and biosorbent regeneration.

The paper might be interesting and useful for researchers in that field and I recommend accepting the paper for publication after major revision.

 1. The Preparation of adsorbent section is not completely clear. The authors wrote that: “The dried algae were cut and then size fractioned by a gravimetrical sieve procedure.” Were the dried algae crushed before sieve procedure? The particles size of the used material should be specified.

2. Biosorbent characterization should be performed, such as FTIR, SEM, and pore distribution and BET surface area of SW before and after the copper adsorption process.

3. Some typo’s and errors should be corrected, for example:

- On pages 4 and 5, Section 2.2.1., instead: “Freundlich Isotherm”, should be: “Freundlich isotherm”.

- In Experimental, page 5, line 183, instead: “Hydrochloric acid” should be: “hydrochloric acid”.

Reviewer 2 Report

Hansen et al., investigated the adsorption of Cu2+ using two brown seaweed species. The adsorption/desorption of pollutants is an interesting and widely investigated topic. The authors should definitely address the novelty of this research, as there are already numerous similar papers to be found in the literature (limiting the added value of this research). The article should also be significantly reorganized to meet the standards of Metals (see comments). The authors should go carefully through the text based on the comments of the reviewer. Overall, the manuscript merits publication in Metals but there are some major and minor comments which ought to be addressed before publication as an extensive revision is needed here.

-     -    The introduction section should be extended significantly as it is currently too generic. The novelty of the research should be addressed properly. Also some important justifications are missing. For example, why is seaweed specifically chosen as the biosorbent (and why specifically brown seaweed species). Is this choice based on availability or for example because of its biochemical components (or both) that influence the adsorption process? It is also advisable to add an overview of the work that is already done in the literature considering Cu2+ adsorption (for instance in table form).

-   -      Section 2 (theoretical background) mainly contains generic textbook information considering the adsorption process. The entire section could be replaced in table form and incorporated in the materials and methods section (in this manuscript experimental). Suggestion would be to combine theoretical background and experimental section into a materials and methods section. Thereby, limiting the theoretical background may be in favor of extending the introduction section (that has more relevant information for the manuscript).

-      -   Did the authors perform a (bio-)chemical analysis on the 2 brown seaweeds? This could reveal important information in the different adsorption phenomena. Just referring to values in the literature is not correct. Especially for seaweed, its composition is highly variable based on geographic location, storage, harvest season,… Also heavy metal concentration could be important here as the seaweed itself can contain considerable amounts of heavy metals (that you don’t want to release via an adsorption process of another metallic compound).

-    -     Is there a specific reason why the authors didn’t apply stirring in their experiments? Did they consider mass transfer limitations?

-      -   The reviewer assumes that the authors were mistaken by addressing a mass to volume ratio of 10 mg/L. Based on Figure 3a and 3b and seaweed amounts mentioned in the manuscript, the reviewer assumes that it should be g/L instead of mg/L?

-      -   The authors apply relatively high biosorbent amounts (up to 40 g/L) for only 10 ppm Cu2+. This is also seen in Table 2 in the retention capacity (that is relatively small). It would be nice to compare this data with values found in the literature and introduce a discussion section.

-     -    Section 4.6, it would be better to refer to a quantitative statistical test than a more qualitative visual inspection.

-       -  There is a typo in Figure 6 “hydrochloric acid”.

-     -    There should be more information added in section 4.7 (related to Figure 6).The authors just mention 0.1 M HCl and H2SO4, but no pH. pH of both solutions is different, and adsorption processes are significantly influenced by pH.

-      -   The quality of the figures should be improved: the grey box surrounding the graphs should be removed. Check for superscripts (for example mg/g-1). Name of seaweed species should be in italic.

-    -     Please go carefully through the text (and tables) and check for significant numbers and be consistent (particle size, q, …).

/

Round 2

Reviewer 1 Report

The authors responded to all my requests and comments, so I recommend accepting the paper for publication in Metals in present form.

Reviewer 2 Report

The authors have addressed the comments raised by the reviewer. Therefore the reviewer accepts this manuscript for publication in Metals.